# Formulation and Evaluation of Polysaccharide Microparticles for the Controlled Release of Propranolol Hydrochloride

**DOI:** 10.3390/pharmaceutics16060788

**Published:** 2024-06-11

**Authors:** Aneta Stojmenovski, Biljana Gatarić, Sonja Vučen, Maja Railić, Veljko Krstonošić, Radovan Kukobat, Maja Mirjanić, Ranko Škrbić, Anđelka Račić

**Affiliations:** 1Centre for Biomedical Research, Faculty of Medicine, University of Banja Luka, Save Mrkalja 16, 78000 Banja Luka, Bosnia and Herzegovina; aneta.stojmenovski@med.unibl.org (A.S.); ranko.skrbic@med.unibl.org (R.Š.); 2Department of Pharmacy, Faculty of Medicine, University of Banja Luka, Save Mrkalja 14, 78000 Banja Luka, Bosnia and Herzegovina; biljana.gataric@med.unibl.org; 3SSPC, The SFI Research Centre for Pharmaceuticals, School of Pharmacy, University College Cork, T12 K8AF Cork, Ireland; svucen@ucc.ie (S.V.); 121100335@umail.ucc.ie (M.R.); 4Department of Pharmacy, Faculty of Medicine, University of Novi Sad, Hajduk Veljkova 3, 21000 Novi Sad, Serbia; veljko.krstonosic@mf.uns.ac.rs; 5Department of Chemical Engineering and Technology, Faculty of Technology, University of Banja Luka, B.V Stepe Stepanovica 73, 78000 Banja Luka, Bosnia and Herzegovina; radovan.kukobat@tf.unibl.org; 6Apoteke Bpharm, Kulska obala bb, 79220 Novi Grad, Bosnia and Herzegovina; maja.mirjanic@student.med.unibl.org; 7Department of Pharmacology, Toxicology and Clinical Pharmacology, Faculty of Medicine, University of Banja Luka, Save Mrkalja 16, 78000 Banja Luka, Bosnia and Herzegovina

**Keywords:** microparticles, propranolol hydrochloride, pediatric population, sodium alginate, hydroxypropyl methylcellulose, hydroxypropyl guar gum, chitosan

## Abstract

Propranolol hydrochloride, a non-cardio-selective beta blocker, is used to treat several conditions in children, including hypertension, arrhythmias, hyperthyroidism, hemangiomas, etc. Commercial liquid formulations are available in Europe and the US, but they have disadvantages, such as limited stability, bitter taste, and the need for multiple daily doses due to the drug’s short half-life. Considering these limitations, controlled-release solid formulations, such as microparticles, may offer a better solution for pediatric administration. The main objective of this study was to formulate an encapsulation system for propranolol hydrochloride, based on sodium alginate and other polysaccharide polymers, to control and prolong its release. Microparticles were prepared using the ionotropic gelation method, which involves instilling a polymer solution into a solution of gelling ions via the extrusion technique. Physicochemical characterization was conducted by assessing the entrapment efficiency, drug loading, swelling index, microparticle size, rheological properties, and surface tension. In order to improve the characteristics of the tested microparticles, selected formulations were coated with chitosan. Further experimental work included differential scanning calorimetry (DSC), Fourier transform infrared (FTIR) analysis, and SEM imaging. This in vitro release study showed that chitosan-coated microparticles demonstrate favorable properties, suggesting a novel approach to formulating pediatric dosage forms, although further optimization is necessary.

## 1. Introduction

It is well established that pediatric patients are not small adults. They have different body compositions and physiological and biochemical processes that govern the pharmacokinetics and pharmacodynamics of drugs [1,2,3]. However, due to the ethical and financial challenges that clinical trials in children bring [4], most of the medicines prescribed for pediatric patients are designed for adults. The pediatric doses are achieved by dilution of liquid formulations, opening of capsules, or fine grinding of tablets, administered as suspensions [5]. This practice is known as off-label drug use, which brings certain risks, such as dosing inaccuracy, unpredictable and variable bioavailability, unknown preparation stability, use of excipients that may be toxic, and a low overall acceptability of the medicine for both children and caregivers [6,7,8]. In order to avoid these risks, pediatric patients must be treated with medications that are developed and approved exclusively for them. The development of pediatric, age-appropriate formulations is a complex process due to numerous factors that must be considered, i.e., taste masking, the heterogeneity of the pediatric population, the requirement to use excipients safe for children, cultural/religious preferences, and the need for dose flexibility [9]. Liquid formulations are the most preferred dosage forms for pediatric patients since they are easy to swallow. However, liquid dosage forms have some serious disadvantages, such as poor stability, which requires the employment of different excipients, such as preservatives, taste-masking problems, and the high cost of storage and transportation [10]. Therefore, there is an agreement to shift from liquid to solid dosage forms, and multi-particulates are considered to be one of the most suitable formulation platforms for oral drug delivery to pediatric patients [8,9,10]. Multi-particulates are small, discrete, multiple-unit dosage forms, including nanoparticles, microparticles, pellets or beads, microcapsules, spheroids, minitablets, etc. These systems are suitable for taste masking by using film-coating technologies [11]. Additionally, dose flexibility, which is often required for pediatric drugs, can be ensured by varying the number or quantity of multi-particulate units administered [9]. It was also reported that multi-particulate formulations were well-accepted and preferred over oral liquids by children from three months of age [12].

Depending on the polymer(s) used in their fabrication, multi-particulates can be programmed for pulsatile, delayed, controlled, or targeted drug release [13]. Pediatric patients with chronic conditions may benefit from controlled-release systems to reduce the frequency of administration [8]. The most utilized polymers in pharmaceutical technology for the preparation of controlled-release drug delivery systems, such as microparticles and nanoparticles, are polysaccharides. This large group of biopolymers are used as encapsulating agents due to their numerous advantages, such as biodegradability, biocompatibility, low toxicity, and swelling properties. Sodium alginate is widely used for the encapsulation of active substances because it has the ability to form a gel structure, such as the “egg-box”, as a result of cross-linking with calcium ions (ionotropic gelation process) [14]. In previous studies, sodium alginate was combined with different polymers, such as hydroxypropyl methylcellulose, polyvinyl alcohol, and gelatin, to formulate dosage forms for pediatric use [15,16].

In this study, sodium alginate was used along with hydroxypropyl methylcellulose and hydroxypropyl guar gum as drug-release modifiers, as well as chitosan as a coating agent, in order to achieve controlled release of propranolol hydrochloride from microparticles formulated for pediatric use. Hydroxypropyl methylcellulose is a water-soluble derivative of cellulose commonly used in drug delivery systems for controlled release [17]. Hydroxypropyl guar gum is the most commonly used derivative of guar gum due to the significantly improved characteristics achieved by the controlled process of etherification of naturally occurring polymer with propylene oxide [18]. Chitosan, as a polycationic polysaccharide and the most important derivative of chitin, has found wide pharmaceutical and drug delivery applications because of its non-toxicity, biodegradability, biocompatibility, non-antigenic nature, and low cost [19]. The thickness of the shell depends on the chitosan molecular weight. Low-molecular-weight chitosan was chosen because it forms a thick shell with a better anti-swelling ability compared to high-molecular-weight chitosan [20].

Propranolol hydrochloride is a non-cardio-selective beta blocker used to treat different conditions in children, such as hypertension, arrhythmias, hyperthyroidism, phaeochromocytoma, hemangiomas, and tetralogy of Fallot, as well as prophylaxis of migraine [21]. Currently, a commercial liquid formulation of propranolol for pediatric patients is authorized in Europe (Hemangiol^®^ 3.75 mg/mL oral solution, Pierre Fabre Medicament Production, Gien, France) and the United States (Hemangeol^®^ 4.28 mg/mL oral solution, Pierre Fabre Pharmaceuticals, Inc., Parsippany, NJ, USA), available on prescription. After first opening, the medicine should be used within two months. Propranolol is rapidly decomposed at alkaline pH and its maximum stability in solution is at pH 3. The commercially available oral solution of propranolol contains citric acid as a stabilizer (pH around 3), sodium saccharin as a sweetener, strawberry and vanilla as flavoring agents to mask the drug’s bitter taste, and propylene glycol (2.6 mg/mL) as a solvent. Daily intake should be limited since they are all labeled as potentially harmful in children and have been associated with toxicity, principally in neonates [22]. According to the provisional pediatric biopharmaceutics classification system (pBCS), propranolol hydrochloride has been reported as one of the drugs with no changes between pBCS and adult BCS. Thus, it is a class I (high permeability and high solubility) compound [23] and is almost completely absorbed after oral administration [24]. However, propranolol is extensively metabolized by the liver and only about 25% of the drug reaches the systemic circulation. It also has a fairly short half-life (3–4 h), which means that, for an optimum effect, the administration of propranolol hydrochloride as an oral solution must be carried out several times a day [24]. Considering all the presented drawbacks of commercially available oral solutions of propranolol, the formulation of a controlled-release solid system, such as microparticles, might be a better solution for the use of this drug in the pediatric population.

Microparticle systems based on polysaccharides, which regulate drug release via diffusion-controlled mechanisms, are described in the literature [25,26,27], but no study has been reported on the encapsulation of propranolol hydrochloride into microparticles intended for the pediatric population. The main objective of this study was to formulate an encapsulation system for propranolol hydrochloride, based on sodium alginate and various other polysaccharide polymers, to control and prolong its release. The method used to develop this system was ionotropic gelation in the presence of calcium chloride as a cross-linking agent. The obtained microparticles were then subjected to different physicochemical characterization, such as particle size measurement, swelling, and dissolution rates in simulated gastric and intestinal media under conditions appropriate for the pediatric population.

## 2. Materials and Methods

### 2.1. Materials

Sodium alginate (Na-ag), the polymer used for the preparation of microparticles, was obtained from Sigma-Aldrich (Oslo, Norway). Hydroxypropyl methylcellulose (HPMC) 4000 mPa·s was procured from Fagron (Nazareth, Belgium). Hydroxypropyl guar gum with molar substitution 0.4 (HPGG) was purchased from Ashland Inc. (Wilmington, DE, USA). Low-molecular-weight chitosan (LCW) with a high degree of deacetylation (DDA = 95%) was purchased from Sigma-Aldrich Chemie GmbH (Taufkirchen, Germany). Propranolol hydrochloride (PCH; manufactured by TCI EUROPE, Zwijndrecht, Belgium) was used as the model drug. Freshly obtained purified water (GenPure apparatus, Thermo Fisher GmbH, Dreieich, Germany) was used for the preparation of polymer dispersions and during the tests. All other reagents and chemicals used were of analytical grade and correspond to the requirements of valid pharmacopoeias (Ph. Eur. and USP).

### 2.2. Preparation of PCH Microparticles

Microparticles were prepared by the method of ionotropic gelation, according to the procedure described by Cadena-Velandia at al. [25], with slight modifications. This method is a chemical method of encapsulation and implies instilling a polymer solution into a solution of gelling ions using the extrusion technique. Different polymer solutions were first prepared by the dispersion of an appropriate amount of Na-ag (1.5–6%, *w*/*w*), alone or in combination with HPMC and HPGG, to 100 mL of distilled water at room temperature for 2 h, under continuous magnetic stirring (2 mag magnetic motion MIX 15 eco, Port Orange, FL, USA) at a speed of 350 revolutions/min (Table 1). Polymer solutions were then refrigerated for 24 h until complete hydration. Then, 4 mL of a 1% PCH solution was added to the Na-ag solutions. The PCH solution was sonicated in an ultrasonicating water bath until completely dissolved. The dispersion was protected from light, and the mixing continued for 1 h, until the drug was thoroughly dispersed. The resulting drug–polymer dispersion was added in drops by the extrusion technique to an aqueous solution of calcium chloride using a syringe with a needle (20 G, 0.9 × 40 mm; Henke Sass Wolf, Tuttlingen, Germany). The concentration of calcium chloride ranged from 1.5 to 6% (*w*/*w*). Formed particles were left in the calcium chloride solution at room temperature for 30 min (hardening process) under gentle agitation. After that, the microparticles were filtered, washed with purified water, and air-dried at 25 °C for 48 h. The compositions of the formulated microparticles are presented in Table 1.

In order to improve the characteristics of the investigated microparticles, selected formulations were coated with chitosan. Preparation of the coated microparticles was carried out in a two-step process, which first included the preparation of microparticles by the procedure already described. The chitosan solution was prepared by dispersing an appropriate amount of polymer in a 0.5% (*v*/*v*) acetic acid solution with mixing on a magnetic mixer (2 mag magnetic motion MIX 15 eco, Port Orange, FL, USA) for 2 h. The chitosan dispersion was left in the refrigerator for 24 h until complete hydration. Adjusting the pH value of the chitosan solution to 5.5 was performed using 0.1 M NaOH. Dry microparticles were then immersed in the chitosan solution (2%, m/m) and stirred with a magnetic stirrer at 350 rpm for one hour. Then, the coated microparticles were filtered, washed with distilled water, and air-dried, as described above [28].

### 2.3. Entrapment Efficiency (%EE) and Drug Loading (%DL) 

The amount of PCH encapsulated in the microparticles was determined by immersing dry microparticles in phosphate buffer at pH 6.8 for 24 h while stirring on a magnetic stirrer (2 mag magnetic motion MIX 15 eco, Port Orange, FL, USA). The mixture containing the microparticles and phosphate buffer was then centrifuged for 15 min at 3000 rpm. Supernatant was filtered through a membrane filter (0.45 µm) (Chromafil^®^ Xtra PTFE-45/25, Macherey-Nagel, Germany). The absorbance at 289 nm was measured with a UV-visible spectrophotometer (UV-1800 spectrophotometer, Shimadzu, Japan) in order to determine the drug concentration. %EE and %DL were calculated as follows:%EE = (C*e*/C*t*) × 100(1)
%DL = (D*e*/D*w*) × 100(2)
where C*t* is the total amount of PCH added to the microparticles during the preparation procedure, C*e* is the amount of PCH inside the microparticles, D*e* is the total amount of PCH encapsulated in the microparticles, and D*w* is the total mass of the microparticles.

### 2.4. Particle Size Measurement

The diameters of dried uncoated and chitosan-coated alginate microparticles containing PCH were determined using a digital micrometer. Particle diameters were recorded in microns. The average particle size was calculated after measuring the diameter of 20 randomly selected particles.

### 2.5. Swelling Study of PCH Microparticles

Determination of the swelling index (SI, %) of dried uncoated and chitosan-coated alginate microparticles containing PCH was performed in two different media: 0.1 M HCl (pH 1.2) and phosphate buffer (pH 6.8). Accurately measured, 100 mg of alginate microparticles was placed in an Erlenmeyer flask containing 50 mL of the appropriate medium, then shaken on an orbital shaker (IKA^®^ KS 260, IKA, Staufen, Germany) for 30 min. Swollen particles were removed within a predetermined time limit and measured after drying the surface with a paper towel. The SI (%) was calculated by the following equation [27]:SI (%) = (W*s* − W*i*)/(W*i*) ×100(3)
where W*s* and W*i* are the mass of the microparticles after swelling and the initial mass of dry microparticles, respectively.

### 2.6. Rheological and Surface Tension Measurements

The rheological behavior of the drug–polymer dispersion was assessed by a HAAKE MARS rheometer (Thermo Scientific, Karlsruhe, Germany) with a cylinder CC25 DIN/Ti sensor. The measurements were performed at a constant temperature of 25 ± 0.1 °C. The flow curves were determined as samples were exposed to a continuous increase of the shear rate from 0.001 to 300 s^−1^ for 120 s. Then, the shear rate was kept constant at 300 s^−1^ for 60 s and, finally, in the following 120 s, it was decreased to 0 s^−1^. Values of the apparent viscosity at the maximum shear rate were used for comparison of the drug–polymer dispersion with different polymer concentrations.

The surface tension of the drug–polymer dispersion was measured on a digital tensiometer, KRÜSS Easy DyneS K20/20023411 (Hamburg, Germany), using the du Noüy ring method at a constant temperature of 25 ± 0.1 °C. All values were automatically corrected using Harkins and Jordan correction factors, integrated into the electronic KRÜSS tensiometer. The ring was immersed in the system and kept for 20 min before starting the measurement. The reported values were the average values of five automatically repeated measurements.

### 2.7. Differential Scanning Calorimetry

Differential scanning calorimetry (DSC) was carried out to determine the thermal behavior and possible interactions between the drug, polymers, and other components in the microparticles. DSC was performed on a Q-1000 DSC (TA Instruments, West Sussex, UK). Samples were accurately weighed into Tzero hermetic aluminum pans, sealed, and heated from 25 °C to 300 °C at a heating rate of 10 °C/min under constant nitrogen flow (50 mL/min).

### 2.8. Fourier Transform Infrared (FTIR) Analysis

FTIR analysis was performed to evaluate the compatibility of PCH with the selected polysaccharide polymers and determine complex formation or potential changes in PCH conformation within the microparticles. Spectra were recorded for samples of PCH, Na-ag, LCW, HPMC, and PCH-loaded chitosan-coated and uncoated microparticles. The measurements were conducted using a Fourier transform infrared spectrophotometer (FTIR, Brucker Tensor 27, Elk Grove Village, IL, USA) equipped with the attenuated total reflectance (ATR) system at 100 scans and a resolution of 4 cm^−1^.

### 2.9. In Vitro Release Study

The release of encapsulated PCH from microparticles was performed using an apparatus with mini paddles (Erweka DT720, Langen, Germany). To mimic pediatric GIT pH conditions, the following dissolution media were changed: 0.1 M HCl (pH 1.2) and phosphate buffer (pH 6.8). The volume of the media was 200 mL each. During the test, the temperature of the media was maintained at 37 ± 5 °C, and the stirring speed at 100 rpm. The test method was adapted to pediatric conditions. The experiment was carried out in two phases. The first phase was in a pH 1.2 medium and lasted 1 h, which imitates the residence time of the drug in the stomach. Then, after changing the medium, the test was continued for another 3 h in a pH 6.8 medium, mimicking conditions in the small intestine. At certain time intervals (after 5, 15, 30, 45, 60, 75, 90, 120, 180, and 240 min), 2 mL of sample was withdrawn and filtered through a membrane filter (0.45 µm). The medium with dissolved microparticles was transferred to the test tubes. The volume of the samples was replaced with the appropriate medium heated to 37 °C. Content of PCH was determined by UV spectrophotometry (UV-1800 Spectrophotometer, Shimadzu, Japan). All the assays were carried out in triplicate. Based on the measured absorbance values at wavelengths of 290 nm (artificial gastric juice, pH 1.2) and 289 nm (phosphate buffer, pH 6.8), the concentration of PCH was determined via calibration curve equations. 

### 2.10. Scanning Electron Microscopy (SEM)

The surface morphology and topographical characteristics of coated microparticles were analyzed by a JEOL JSM 5510 Scanning Electron Microscope (JEOL Ltd., Tokyo, Japan). Samples were mounted onto aluminum stubs using double-sided carbon tape. All samples were sputter-coated with a 5 nm layer of gold palladium (80:20) using a Quorum Q150 RES Sputter Coating System (Quorum Technologies, Lewes, UK), before being examined using the JEOL SEM instrument. Digital electron micrographs were obtained of areas of interest. The particle sizes were measured from many images. The particles with the same diameter were counted, and the fractions of the particles (in percent) with the same diameter are shown in histograms.

## 3. Results and Discussion

### 3.1. Preparation of PCH Microparticles

A total of eight formulations were prepared, varying the amounts of Na-ag and calcium chloride (1:1), according to the procedure described in the Methods Section. Microparticles were also prepared using polymers mixed with Na-ag, such as HPMC (F6) and HPGG (F7 and F8), in two different concentrations, as shown in Table 1. All the prepared formulations were subjected to characterization for entrapment efficiency, drug loading, and size measurement.

### 3.2. Entrapment Efficiency (%EE) and Drug Loading (%DL)

The encapsulation efficiency values of all tested formulations ranged between 21.30 and 65.00% (Table 2). To investigate when the drug was lost in more detail, the calcium chloride solution was subjected to spectrophotometric analysis after filtration. Based on these findings, it can be concluded that the drug diffusion occurred to the greatest extent during the first preparation step, although the ionotropic gelation reaction was fast [29]. Different concentrations of Na-ag and calcium chloride were used to obtain a higher EE (%). The entrapment efficiency of the formulations with a single polymer (F1 to F5) was found to be in the range of 21.3 ± 1.6 to 54.9 ± 1.2%, with formulation F2 (2% (*w*/*w*) of Na-ag) possessing the highest entrapment efficiency. Various polymers were used with Na-ag to modulate drug loading and modify the drug-release kinetics [30]. Adding HPMC and HPGG helped with the effective encapsulation of the drug in microparticles. Among the investigated formulations, the one that showed the highest EE % (F6) was prepared by mixing HPMC with Na-ag dispersion.

Drug loading was found to be low and in the range of 0.12 ± 0.01 to 0.27 ± 0.02% in all formulations. Similar results have been shown in the literature [30,31]. The low drug loading can be explained by the high porosity of the alginate gel matrix, which enables the dissolution of PCH in water during the formation and recovery of microparticles. The lowest DL was recorded for the formulation with Na-ag concentrations above 2%. Low DL may also be associated with a small initial amount of the drug used to prepare the microparticles [30]. 

Microparticles that demonstrated the highest drug EE (%) and DL (%) were identified and described in further studies. Additionally, these formulations were coated with acidic chitosan dispersion.

### 3.3. Microparticle Size Measurement

The results of the mean microparticle size measured by the digital micrometer are presented in Table 2. The size of microparticles ranged between 641 ± 7 and 821 ± 23 μm, which confirmed the particle sizes in the micron range. It is evident that with the increasing concentration of Na-ag, the sizes increased. A similar trend was observed with the addition of HPMC and HPGG. Variations in the particle size with changes in the polymer concentration could be related to an increase in the viscosity of the polymer dispersions, which led to the formation of large droplets during the addition of the drug–polymer dispersion into the calcium chloride solution [32,33].

Particle size measurement was also performed after coating the selected formulations (FC2 and FC6), and the mean values were 761 µm and 884 µm, respectively. The results indicated that the coated formulations had larger particle sizes when compared to the uncoated formulations. The slight increase in the diameter of the coated particles indicates that the chitosan remained on the surface of the microparticles in the form of a thin film and that the coating process was successful [30].

### 3.4. Swelling Study of PCH Microparticles

A swelling study was performed with uncoated and coated formulations for a better understanding of drug release that is influenced by the swelling of microparticles [32]. The swelling phenomenon refers to the absorption of water by the microparticle matrix in order to fill the dehydrated regions that arise due to the relaxation of the polymer network [34]. The results of the swelling study are presented in Table 3. Na-ag is a polymer that exhibits pH-dependent behavior and is known to be almost insoluble in acidic media. The degree of swelling depends on the pH value of the medium in which the microparticles were tested [30]. At pH 1.2, it swelled minimally, while at pH 6.8, the polymer chains relaxed and the microparticles received larger amounts of medium, which explains the extremely high swelling index at this pH value. The small swelling index of Ca-alginate microparticles in the acidic environment was a consequence of proton–calcium ion exchange, which led to the formation of insoluble regions of alginate during the penetration of the medium into the dense polymer network. The swelling behavior of uncoated microparticles was also related to the buffer composition. In phosphate buffer with pH 6.8, exchange of Ca^2+^ ions from the polymer matrix with Na^+^ ions from the phosphate buffers and formation of calcium phosphate was expected. Based on the above, it can be concluded that microparticles slightly swell in the stomach, while the majority of the swelling and erosion processes take place in the small intestine [34].

By introducing HPMC into the microparticle formulation, there was a notable increase in the swelling index under both acidic (pH 1.2) and neutral to basic (pH 6.8) conditions. This growth is explained by the hydration and subsequent swelling of HPMC within the microparticles upon exposure to the medium. The heightened swelling index was attributed to the larger hydrodynamic volume occupied by the polymer chains of HPMC, in contrast to microparticles containing only Na-ag. The incorporation of HPMC significantly enhanced the swelling index of the microparticles across varying pH levels, highlighting its role in magnifying the overall swelling capacity of the formulation. These results are in agreement with those found by Yahoum et al. [27], who reported similar results for microparticles based on xanthan gum and linked this to the presence of more pores and cavities on the microspheres. Increasing the concentration of polysaccharide polymers in the formulation of microparticles significantly affected the degree of swelling, which consequently led to changes in the drug-release behavior [33].

There were significant changes compared to the results obtained for uncoated microparticles when the swelling index was determined for coated microparticles (Table 3). The swelling behavior of coated microparticles was mainly determined by the properties of the coating polymer, but after its dissolution, the swelling was affected by the properties of the Ca-alginate matrix, i.e., HPMC–Ca-alginate. Both coated formulations retained a higher swelling index in the medium with pH 6.8 compared to the acidic medium. However, there was a noticeable decrease in the swelling index and a reduced water uptake capacity after coating the microparticles. The less pronounced cationic nature of chitosan under neutral and slightly basic conditions (pKa of chitosan is 6.3) led to a lower binding of chitosan to alginate, which affected the swelling [34].

Chitosan was used as a membrane-coating material in order to improve the stability of alginate microparticles in the gastric environment and prolong the release of the drug. The decrease in the swelling index in the pH 6.8 medium indicates that chitosan slowed down the swelling and disintegration of the microparticles.

### 3.5. Rheological and Surface Tension Measurements

The physicochemical characterization of drug–polymer dispersions, marked with P1–P8, which were used for the production of microparticles, included rheological characterization and measurement of surface tension. The viscosity of the polymer dispersion significantly affected the possibility of extrusion through the syringe, the droplet size, and consequently, the shape and size of the formed microparticles. Highly viscous polymer solutions are usually not used due to nozzle flow problems, such as plugging and no flow. When the liquid exits through a small opening, it experiences different stresses; firstly, the influence of surface tension, but also shear stress [35]. The apparent viscosity of drug–polymer dispersions was measured at a shear rate ranging from 0.001 to 300 s^−1^, with a 60 s hold at the maximum shear rate, because the applied force during extrusion of the dispersion through the syringe was also constant. Since there was no significant change in viscosity during this time, the mean value of the apparent viscosity is shown in Table 4. The viscosity of the polymer solution affected its ability to resist flow, which mainly depended on the hydrodynamic volume of polymer chains. The conformation and flexibility of the polymer chains were influenced by the concentration of the polymer and the average length of the chains, but also experimental conditions (temperature, pH, solvent, and ionic strength) [27]. It was observed that the apparent viscosity values increased as a function of the Na-ag concentration (P1–P5). The influence of an additional polymer (HPMC or HPGG) on apparent viscosity was also evident (P6–P8).

The higher the polymer concentration, the greater the entanglement of the polysaccharide chains, which means that the resulting gel structure was more rigid and created greater resistance to flow, which consequently made extrusion of polymer dispersion through the needle more difficult [27]. The production of microparticles with polymer dispersions of extremely high viscosity (>500 mPa·s) was difficult, and particles of a deformed shape were obtained. On the other hand, it was confirmed that the viscosity of the polymer solution must be above 60 mPa·s to produce microparticles with the desired spherical shape and good mechanical properties [36]. The microparticles’ size directly depends on the viscosity properties of the polymer dispersion [36]. The two selected microparticle formulations (F2 and F6) were prepared using polymer dispersions whose viscosity was 73.74 ± 0.33 and 171.97 ± 0.81, respectively. The apparent viscosity curves of these drug–polymer dispersions (Figure 1) indicated a non-Newtonian character and pseudoplastic behavior, characterized by the decreasing viscosity with the increasing shear rate. It can be noticed that the non-Newtonian behavior was more pronounced in the dispersion P6, to which, in addition to Na-ag, one more polysaccharide polymer (HPMC) was added. Previous studies reported that alginate dispersions exhibited a non-Newtonian character, which can affect the formation of droplets [36].

The surface tension of polymer dispersions was in the range from 43.0 ± 0.09 to 45.4 ± 0.31, which is in agreement with other reported results for alginate [37]. Taking into account that the surface tension of water is higher and according to literature data is 72 mN/m [36], the presence of polymers in water caused a certain surface activity. However, previous literature data suggested that static measurements of surface tension are probably not representative indicators of behavior during droplet formation. Additionally, the hydrocolloid properties of alginates were highly dependent on the ability to hydrate them to form viscous solutions through dispersions or the gelation process [36,37,38].

### 3.6. Differential Scanning Calorimetry

The DSC thermograms of PCH, CaCl_2_, Na-ag, HPMC, and LCW are depicted in Figure 2. PRO exhibited a sharp melting endothermic peak at 167.03 °C [39]. Thermograms of LCW and Na-ag showed an endothermic peak followed by an exothermic peak. A broad endothermic peak of Na-ag at about 154 °C may indicate water evaporation from the polymer. Loss of water at such a high temperature can be explained by the presence of strong ion-dipole bonds between –COO– groups of Na-ag and water molecules [29]. Further, a sharp exothermic peak was observed at 230 °C, probably due to polymer decomposition. The chitosan thermogram showed a broad endotherm peak with a maximum at about 178 °C, followed by a broad degradation exothermic event with the maximum at 283 °C. DSC thermograms of CaCl_2_ exhibited two sharp endothermic peaks at 178.67 °C and 267.10 °C. HPMC showed a broad endothermic peak at about 130 °C due to the loss of water or adsorbed moisture from the polymer. These findings are in line with previously reported data from DSC analyses [29,34,40].

Regarding the physical mixtures of the drug and polymers (FM2), the thermograms showed three endothermic peaks (Figure 3a), one correlated with the drug (148.60 °C) and two with Na-ag and CaCl_2_. A similar thermogram was obtained from the physical mixture with LCW (FMC2), with one sharp peak at 146.43 °C. On the other hand, for the uncoated microparticles, there was only one endothermic peak at 218.35 °C, which probably resulted from the melting of the microparticles. The thermograms of both the physical mixture (FM2) and the microparticles (F2) showed exothermic peaks at 209.04 °C and 192.57 °C, which correspond to Na-ag. This peak remained on the thermogram of the physical mixture with LCW (FMC2) and was significantly reduced in coated microparticles (FC2). These changes in thermal characteristics indicate the formation of a polymeric complex due to the presence of strong ionic interactions [29]. Physical mixtures of PCH with other excipients induced shifting of the PCH endotherm, corresponding to its melting point, at 19–20 °C or lower, compared to PCH alone. In contrast, the absence of a PRO endothermic peak in the microparticle thermograms suggests that PCH was completely entrapped in the polymer matrix or, probably, that its crystals transformed to an amorphous form. The transition of the drug into an amorphous form indicates better solubility and bioavailability [34,39]. 

The thermograms shown in Figure 3b represent the thermal behavior of the drug and polymers in the second physical mixture containing PCH/CaCl_2_/Na-ag/HPMC (FM6) and their physical mixture with LCW (FMC6), uncoated (F6), and coated (FC6) microparticles. The endothermic peak present in both physical mixtures at 147.73 °C and 151.13 °C, respectively, probably corresponds to the melting of the drug. A shift of the peak toward lower temperatures was observed, which indicated the possibility of an interaction of PCH with the polymers. The thermogram of uncoated and coated microparticles (F6 and FC6) did not show the melting peak of the drug, indicating the amorphization of PCH [39]. 

### 3.7. Fourier Transform Infrared (FTIR) Analysis

FTIR spectra were recorded to confirm the results obtained by DSC analysis. The possible drug–polymer interactions were investigated by comparing the FTIR spectra of coated and uncoated microparticles with spectra of the pure drug and polymers. The FTIR spectra of PCH, Na-ag, and LCW were consistent with previous reports (Figure 4).

PCH exhibited several absorption bands near 3277 and 2965 cm^−1^ from the hydroxyl group (secondary) and secondary amine group, and a stretching band at 1105 cm^−1^ corresponding to C–O–C stretching in aryl alkyl ether. The peak at 797 cm^−1^ originated from α-substituted naphthalene [41,42]. The obtained spectrum of Na-ag showed absorption bands at around 3390 cm^−1^, typical for –O–H stretching vibrations, a sharp band between 1627 and 1589 cm^−1^ attributed to C=O stretching vibrations, a band at 1402 cm^−1^ that originated from carboxylate anion, i.e., C–O–C stretching vibrations, and bands at 1083–1024 cm^−1^ corresponding to O–H twisting vibration of carboxylic acid and C–O stretching vibrations [25,26]. In the FTIR spectrum of LCW, the absorption band at 3360 cm^−1^ originated from the stretching of –O–H groups, and that at 2873 cm^−1^ was typical of –C–H stretching vibrations. Amino groups of LCW showed broad absorption bands at around 1654 cm^−1^ corresponding to C=O stretching (amide I), at 1552 cm^−1^ from –N–H vibrations (amide II), and at around 1377 cm^−1^ from amide III. These absorption bands are characteristic of chitosan and are in agreement with previously published results [26,43]. In another formulation (F6), additional pure polymer HPMC showed a broad band at 3460 cm^−1^ due to –O–H stretching vibration and intermolecular H- bonding. The broad band between 2935 and 2841 cm^−1^ was attributed to stretching vibration of –C–H, and the band at 1645 cm^−1^ indicated the presence of C=O stretching of the carbonyl group. The sharp peak at 1049 cm^−1^ originated from stretching vibration of –C–O groups [44,45].

The FTIR spectra of the physical mixtures containing PCH and the excipients, used at a ratio of 1:1 (FM2 and FM6), showed the most characteristic strong absorption bands of the drug and polymers, which were located in almost identical positions in the spectra of pure substances. These results confirmed the compatibility between the drug and excipients used for the preparation and coating of microparticles. In Figure 4, the FTIR spectra of the formulated coated microparticles (FC2 and FC6) also demonstrate the absence of incompatibility between PCH and other microparticle components.

### 3.8. In Vitro Release Study

The drug release from polymer microparticles depends on penetration of the dissolution medium into microparticles, the swelling index and dissolution of the polymeric matrix, and dissolution of the drug after washing through the swollen matrix [30]. The in vitro release experiments of PCH microparticles were performed in two phases: first in simulated gastric medium (pH 1.2) and second in intestinal medium (pH 6.8). The results of the in vitro release study of PCH as a function of time from the uncoated and coated microparticles are presented in Figure 5a,b. It can be observed that 65.5 ± 3.4% of the drug was released from formulation F2 after only 5 min. Furthermore, after 15 min, 84.3 ± 2.0% of PCH was released. Complete release, i.e., 98.3 ± 5.3%, occurred after 90 min. 

Regarding the microparticles with the added HPMC copolymer (formulation F6), the initial release rate was slower compared to F2. Specifically, after 5 min, approximately 36.2 ± 1.99%, and after 15 min, about 75.3 ± 3.3% of the drug was released. Additionally, the complete release also took more time compared to F2, with approximately 101.2 ± 1.5% of PCH being released after 120 min. This is in agreement with previous literature data. When Na-ag was combined with other polymers, such as CMC, chitosan, pectin, and various other polymers, it was found that the gel strength and drug-release properties were enhanced when compared to microparticles prepared using Na-ag alone [30,46]. The Na-ag and HPMC mixture formed a more rigid gel, which decreased the calcium ion diffusion from the cross-linked calcium alginate matrix and reduced gel erosion [30]. Additionally, the drug–polymer dispersion (P6) with HPMC presented a higher viscosity of 171.97 ± 0.81 mPa·s and, as expected, a higher average size. According to the literature data and the relationship between size and surface area, larger particles presented a smaller surface area. The larger microparticles (F6) were able to protect the drug in a higher extension due to the smaller surface area than the smaller microparticles. Regardless of all the above, combining Na-ag with HPMC did not provide the desired prolonged drug release from formulation F6, only taking 60 min.

By encapsulating PCH in microparticles, its release would be expected after 60 min. Furthermore, when the pH 1.2 medium was replaced by the pH 6.8 medium, which mimics the passage of microparticles from the stomach to the small intestine, Na-ag showed a pH-dependent behavior, in the sense that in an acidic environment (pH 1.2), it swelled minimally, while at pH 6.8, the polymer relaxed, swelled, and consequently released the encapsulated drug. Since the encapsulation of the drug did not occur, but adsorption was achieved probably due to the presence of nicks on the surface of the microparticle beads, the release of PCH occurred much earlier at pH 1.2. This was also confirmed by the fact that PCH itself is highly soluble at low pH values [30]. According to these results, the investigated microparticles can be used as useful carriers for PCH delivery, but the programmed release in the small intestine, which was the goal of the research, was not achieved.

In order to achieve the programmed release of PCH, the microparticles of formulations F2 and F6 were coated with chitosan. After coating, the in vitro dissolution rate of PCH from formulations FC2 and FC6 was tested again. As displayed in Figure 5b, formulation FC2 released 17.5 ± 0.4% in the first 5 min. This value increased to 24.7 ± 0.3% in the first hour of testing at pH 1.2. In the second phase of the dissolution test, at pH 6.8, after 120 min, 31.3 ± 1.5% of the drug was released. At the end of the experiment, i.e., after 240 min, the percentage of released drug was 45.6 ± 1.0%. On the other hand, the formulation containing the polymer combination (FC6) again showed improved drug-release properties. In the first phase of the test at pH 1.2, 31.4 ± 0.3% of the drug was released within one hour. After the medium was changed, the percentage of released drug gradually increased, reaching 81.9 ± 3.0% after 240 min. These results are consistent with the swelling study, showing that the FC6 formulation swelled more and released a higher amount of the drug.

Hence, it can be observed that a significant portion of the encapsulated PCH may be released in the intestine rather than the gastric environment. The release rate of PCH was much more prolonged in case of chitosan-coated microparticles, indicating its protective efficiency in the gastric environment and achieving controlled release of the drug. Angadi et al. [32] compared in vitro release of amoxicillin from chitosan-coated and uncoated formulations and concluded that enteric coating of sodium alginate composite beads with chitosan effectively slowed down the release of amoxicillin in the gastric medium. Similar to these findings, in another study, magnetic alginate coated with chitosan was synthesized for controlling the release of amoxicillin. Coating the microparticles with chitosan in this case led to a two-fold increase in the release time. The drug was mainly released in the simulated intestine solution, indicating that the microparticles tend to release the drug in a higher pH environment, rather than in the acidic environment of the stomach [47]. Chitosan was also used for coating of pH-responsive porous silicon to ensure controlled delivery of lutein. An in vitro release study showed a pH-responsive and controlled release of lutein within 7 h [48].

### 3.9. Scanning Electron Microscopy (SEM)

Based on the previous results, coated FC2 and FC6 microparticles were chosen for SEM analysis. Scanning electron micrographs of coated microparticles containing Na-ag alone are shown in Figure 6a–c, while the morphology of the microparticles with Na-ag and HPMC is presented in Figure 6d–f. There were some differences in the shape and surface microparticle morphology but, generally, a spherical morphology of FC2 microparticles could be observed. In contrast, the FC6 microparticles were found to be predominantly ellipsoidal in shape, probably due to the higher viscosity of the polymer dispersion and the way the droplet was formed. Figure 6c,f illustrate that both microparticles possessed a porous surface characterized by numerous wrinkles, likely resulting from the partial collapse of the polymer network during the drying process. Figure 6b,e depict FC2 and FC6 microparticles with dimensions that correspond to those previously measured (Table 2).

It has already been reported that Ca-alginate microparticles coated with chitosan usually have a complex structure with a loose core and a dense surface layer due to different gelation mechanisms, which leads to the collapse of microparticles during the drying process [32]. The pronounced roughness and porosity on the surface that occurred during drying indicate that a large amount of water was retained during the coating of microparticles with chitosan [30]. As the microparticles dried, the retained water was released through the highly porous polymer network, forming the pits in the regions of high porosity. This led to the collapse of the porous gel structure. Also, the agglomeration tendency of the chitosan coating at the microparticles’ surface may be related to the adhesive nature of chitosan [30,49]. SEM results of coated microparticles showed the importance of the drying process for morphology and surface properties of microparticles. Changes in the spherical shape can also be caused by the adhesive nature of chitosan and the type of substrate on which the drying process was carried out [29,30]. However, the presence of porous surfaces in coated microparticles had a significant effect on water retention and uptake of physiological media, which consequently improved swelling and affected drug release [50].

The particle size distribution histograms with shorter and longer dimensions were derived from the SEM images (Figure 7). Microparticles had an elliptical shape with a relatively narrow size distribution on the microscale. As emphasized earlier, the FC6 microparticles were larger than the FC2 microparticles in shorter and longer directions due to aggregation of the polymers during particle formation. The addition of HPMC to the formulation increased the viscosity, suggesting an increase in the particle size [40]. Thus, HPMC stimulated the particle growth, resulting in large micron-to-millimeter-order particles.

## 4. Conclusions

The ionotropic gelation technique is a simple method that can be successfully used for preparation of propranolol hydrochloride microparticles using sodium alginate and other polysaccharide polymers, such as HPMC and HPGG, as drug-release modifiers. Polymer dispersions with different combinations and concentrations of polymers were investigated for their viscosity and surface tension in order to assess their influence on the shape and size of the produced microparticles. The obtained particles were also characterized for their particle size, entrapment efficiency, drug loading, swelling behavior, and surface morphology. The interaction study between the drug and polymers conducted using Fourier transform infrared (FTIR) spectroscopy ruled out any possible incompatibility. Significant differences in the in vitro drug release were observed between formulations coated using chitosan compared to uncoated microparticles. Ca-alginate microparticles coated with chitosan swelled at pH 1.2 but underwent diffusion and erosion at pH 6.8. This study showed that chitosan-coated microparticles demonstrated favorable properties, suggesting a novel approach to formulating pediatric dosage forms, although further optimization is necessary. The coating with chitosan significantly retarded drug release from alginate/HPMC microparticles. Further evaluation using in vivo models should be performed to confirm the in vitro results.

## Figures and Tables

**Figure 1 pharmaceutics-16-00788-f001:**
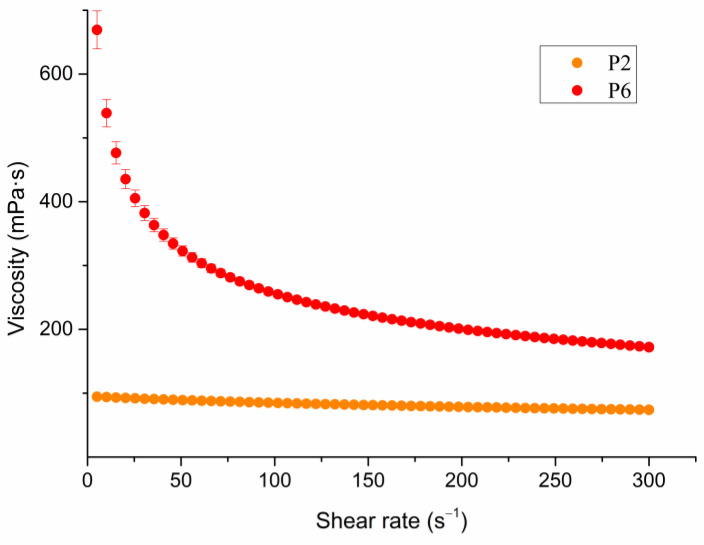
Apparent viscosity versus shear rate curves for selected drug–polymer dispersions.

**Figure 2 pharmaceutics-16-00788-f002:**
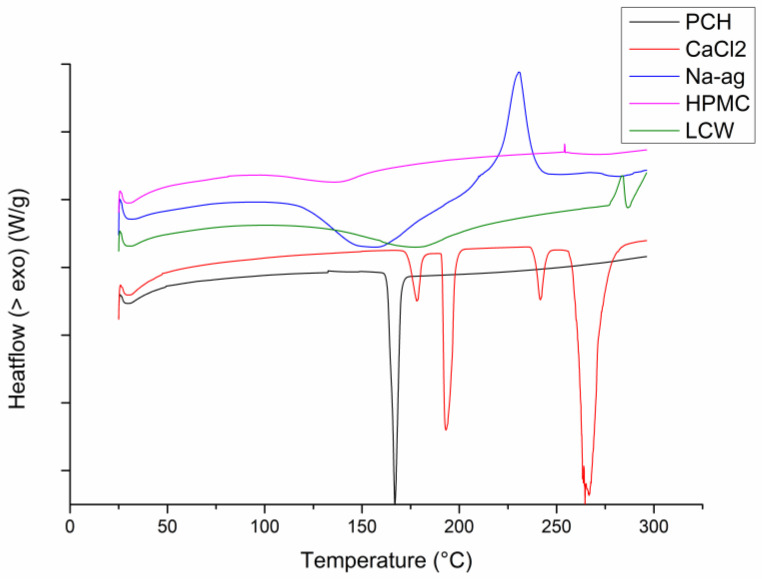
DSC thermograms of the drug (PCH), inotropic agent (CaCl_2_), and polymers used.

**Figure 3 pharmaceutics-16-00788-f003:**
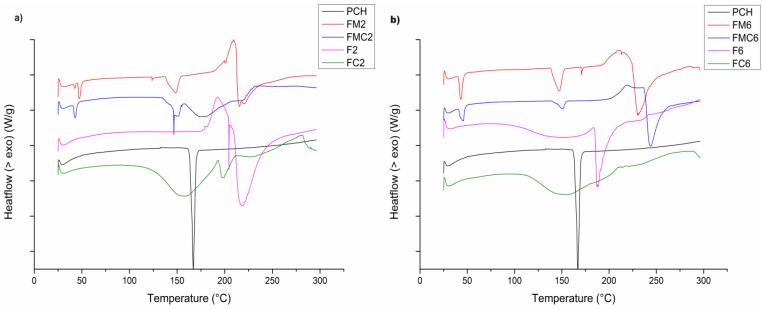
DSC thermograms of (**a**) PCH, physical mixture of PCH/CaCl_2_/Na-ag (FM2), physical mixture of PCH/CaCl_2_/Na-ag/LCW (FMC2), uncoated (F2), and coated (FC2) microparticles. (**b**) PCH, physical mixture of PCH/CaCl_2_/Na-ag/HPMC (FM6), PCH/CaCl_2_/Na-ag/HPMC/LCW (FMC6), uncoated (F6), and coated (FC6) microparticles.

**Figure 4 pharmaceutics-16-00788-f004:**
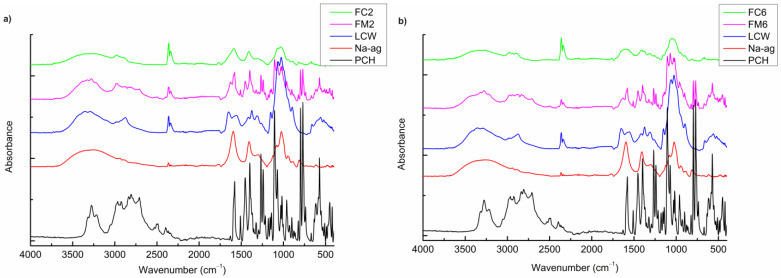
FTIR spectra of (**a**) PCH, Na-ag, LCW, physical mixture of PCH/CaCl_2_/Na-ag (FM2), and coated microparticles (FC2), and (**b**) PCH, HPMC, Na-ag, LCW, physical mixture of PCH/CaCl_2_/Na-ag/HPMC (FM6), and coated microparticles (FC6).

**Figure 5 pharmaceutics-16-00788-f005:**
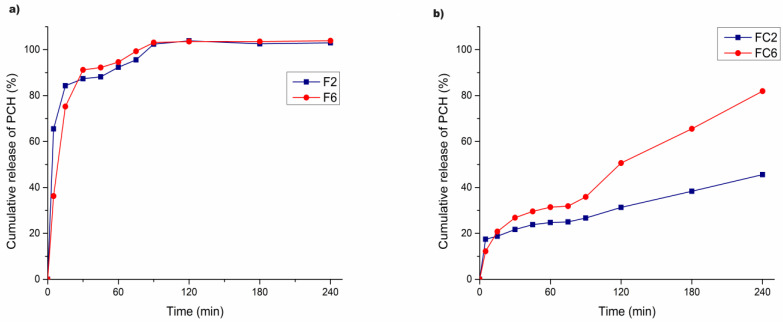
In vitro drug release from (**a**) uncoated and (**b**) chitosan-coated microparticles.

**Figure 6 pharmaceutics-16-00788-f006:**
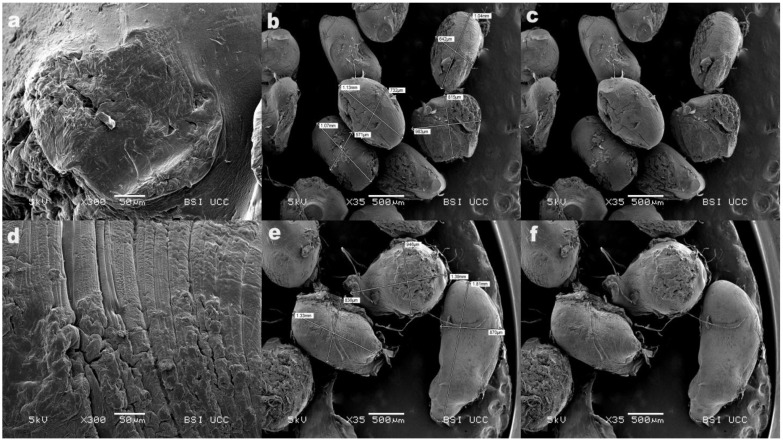
SEM micrographs of (**a**) FC2 microparticles (300×), (**b**) FC2 microparticles with dimensions (35×), (**c**) FC2 microparticles (35×), (**d**) FC6 microparticles (300×), (**e**) FC6 microparticles with dimensions (35×), and (**f**) FC6 microparticles (35×).

**Figure 7 pharmaceutics-16-00788-f007:**
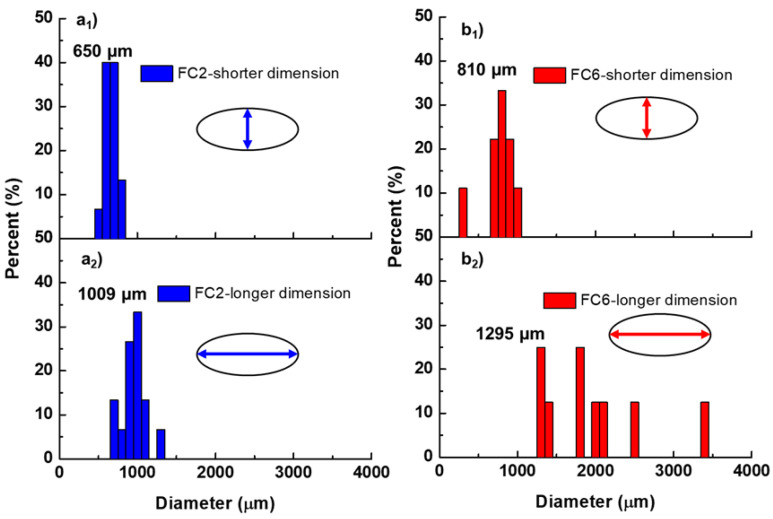
Microparticle size distribution histograms for longer and shorter dimensions, as shown in the inserts. (**a_1_**) FC2 microparticles measured at the shorter dimension. (**a_2_**) FC2 microparticles measured at the longer dimension. (**b_1_**) FC6 microparticles measured at the shorter dimension. (**b_2_**) FC6 microparticles measured at the longer dimension.

**Table 1 pharmaceutics-16-00788-t001:** Composition of Na-ag microparticles.

Formulation Code	Na-ag (%, *w*/*w*)	Calcium Chloride (%, *w*/*w*)	HPMC (%, *w*/*w*)	HPGG (%, *w*/*w*)
F1	1.5	1.5	-	-
F2	2	2	-	-
F3	4	4	-	-
F4	5	5	-	-
F5	6	6	-	-
F6	2	2	1	-
F7	2	2	-	0.25
F8	2	2		0.5

**Table 2 pharmaceutics-16-00788-t002:** EE (%) and DL (%) of PCH microparticles.

Formulation Code	Size (μm) *	EE (%) ^#^	DL (%) ^#^
F1	641 ± 7	21.3 ± 1.6	0.18 ± 0.02
F2	670 ± 10	54.9 ± 1.2	0.27 ± 0.02
F3	691 ± 19	31.5 ± 1.5	0.12 ± 0.01
F4	787 ± 12	47.2 ± 1.8	0.13 ± 0.03
F5	821 ± 23	51.0 ± 2.5	0.13 ± 0.01
F6	698 ± 12	65.0 ± 1.5	0.25 ± 0.01
F7	710 ± 25	55.8 ± 2.4	0.22 ± 0.02
F8	741 ± 33	49.1 ± 3.8	0.20 ± 0.02

* Mean ± SD, *n* = 20. ^#^ Mean ± SD, *n* = 3.

**Table 3 pharmaceutics-16-00788-t003:** Swelling index (%).

Formulation Code	pH 1.2	pH 6.8
F2	71.6	399.3
F6	148.8	626.5
FC2 *	98.05	278.23
FC6 *	135.04	335.59

* Coated microparticles.

**Table 4 pharmaceutics-16-00788-t004:** The apparent viscosity values of drug–polymer dispersions.

Polymer(s) Dispersion	Na-ag (%, *w*/*w*)	HPMC (%, *w*/*w*)	HPGG (%, *w*/*w*)	Viscosity (mPa·s) ^1^
P1	1.5	-	-	37.46 ± 0.15
P2	2	-	-	73.74 ± 0.33
P3	4	-	-	212.97 ± 1.56
P4	5	-	-	802.70 ± 1.73
P5	6	-	-	1328.33 ± 4.04
P6	2	1	-	171.97 ± 0.81
P7	2	-	0.25	287.07 ± 0.21
P8	2	-	0.5	481.70 ± 2.31

^1^ Viscosity value at a shear rate of 300 s^−1^.

## Data Availability

The data presented in this study are available in this article.

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
