# Peer review of "Formulation and Evaluation of Polysaccharide Microparticles for the Controlled Release of Propranolol Hydrochloride"

_pharmaceutics, 2024, doi:10.3390/pharmaceutics16060788_

Round 1
Reviewer 1 Report
Comments and Suggestions for Authors
The authors report an encapsulation system for propranolol hydrochloride to control and prolong its release. Microparticles were prepared using ionotropic gelation method which involves instilling a polymer solution into a solution of gelling ions via the extrusion technique. Physicochemical characterization was conducted by many methods. Chitosan coated microparticles demonstrate favorable properties. The results are very interesting. However, some points of the manuscript should be improved. Specific comments are given below.
1. The structure of encapsulation system should be further measured by TEM.
2. The hydrodynamic size and zeta potential should be measured.
3. The Figure 5. Should be added the standard deviation.
4. The cytotoxicity of encapsulation system should be measured.
5. The in vitro drug release from a) uncoated and b) chitosan coated microparticles should be compared with other reports.
6. More recent published reference should be added in this manuscript.
Comments on the Quality of English LanguageMinor editing of English language required
Reviewer 2 Report
Comments and Suggestions for Authors
Point 1:- As mentioned in line 506-618, that difference between F2 and F6 release is hardly around 60 minutes , is it signicfant factor time.
Point 2:- in fiigure 6d, the picture represent scaling of microparticles as they have many layers of coating which had been like threads layers rather than smooth coating
Point 3:- invitro release was done on pH1.2 and pH6.8 , do similar effect can be seen in stimulated gastric fluid.
Point 4:- in figure 5, no std deviation used and whether it was done in triplicate or single time
Point 5:- in table 4, as you had used two different conc of chitosan but in case of HPMC only one concentration used, was it predetermined or randomly done. Did DOE needed to be applied
Point 6:- Do viscosity play important role in release kinetics and how it was interpreted
Point 7:- in figure 1, what do you mean by P2 and P6 if they denotes formulation then denotes same for all over the manuscirpt
Point 8:- in figure 5, whether the release mention in pH1.2 or pH 6.8, use different colors for pH based or differentiate them in graph
Round 2
Reviewer 1 Report
Comments and Suggestions for Authors
The authors have addressed the problem very well, and the manuscript can be accepted in the present form.
Comments on the Quality of English LanguageMinor editing of English language required
Author Response
We sincerely thank the Reviewer #1 for this comment.
The manuscript was read by a native speaker and minor English language corrections were made.
